# The Effects of Statin Treatment on Serum Ferritin Levels: A Systematic Review and Meta-Analysis

**DOI:** 10.3390/jcm11175251

**Published:** 2022-09-05

**Authors:** Tannaz Jamialahmadi, Mitra Abbasifard, Željko Reiner, Manfredi Rizzo, Ali H. Eid, Amirhossein Sahebkar

**Affiliations:** 1Applied Biomedical Research Center, Mashhad University of Medical Sciences, Mashhad 9177948564, Iran; 2Surgical Oncology Research Center, Mashhad University of Medical Sciences, Mashhad 9177948954, Iran; 3Immunology of Infectious Diseases Research Center, Research Institute of Basic Medical Sciences, Rafsanjan University of Medical Sciences, Rafsanjan 7717933777, Iran; 4Department of Internal Medicine, Ali-Ibn Abi-Talib Hospital, School of Medicine, Rafsanjan University of Medical Sciences, Rafsanjan 7717933777, Iran; 5Department of Internal Medicine, University Hospital Center Zagreb, 10000 Zagreb, Croatia; 6Department of Health Promotion, Mother and Child Care, Internal Medicine and Medical Specialties, School of Medicine, University of Palermo, 90133 Palermo, Italy; 7Department of Basic Medical Sciences, College of Medicine, QU Health, Qatar University, Doha P.O. Box 2713, Qatar; 8Biotechnology Research Center, Pharmaceutical Technology Institute, Mashhad University of Medical Sciences, Mashhad 9177948954, Iran; 9Department of Biotechnology, School of Pharmacy, Mashhad University of Medical Sciences, Mashhad 9177948954, Iran

**Keywords:** acute-phase response, lipid-lowering therapy, inflammation, iron, statins, cardiovascular diseases

## Abstract

Background: Statins are the most widely used drugs for decreasing elevated serum LDL-cholesterol (LDL-C) and thus for the prevention of atherosclerotic cardiovascular disease (ASCVD), but they have also some pleiotropic effects, including anti-inflammatory properties. Atherosclerosis is a low-grade inflammatory disease, and elevated ferritin is considered to be one of the markers of inflammation. Since the results of studies on the effects of statins on serum ferritin levels are conflicting, this meta-analysis was performed. Methods: A literature search was performed using major electronic databases (MEDLINE/PubMed, Scopus, Embase, and ISI Web of Science) from inception up to 5 March 2022 to find studies evaluating the effect of different statins on serum ferritin levels. The effect size was determined using weighted mean differences (WMDs) and the corresponding 95% confidence intervals (CIs). Results: The meta-analysis of nine studies (1611 patients) analyzing the effects of statins on serum ferritin levels that were included showed a significant decrease in circulating ferritin levels caused by statins. The results did not suggest any significant association between the changes in concentrations of serum ferritin and the duration of treatment with statins. Conclusions: Statin therapy decreases the circulating concentrations of ferritin, which might be beneficial for the prevention and/or progression of ASCVD. This effect might be explained by the anti-inflammatory effects and maybe some other pleiotropic effects of statins and not by their lipid-lowering effects.

## 1. Introduction

Statins are inhibitors of 3-hydroxy-3-methylglutaryl coenzyme A (HMG-CoA) reductase, an enzyme crucial for the synthesis of cholesterol in cells, particularly hepatocytes. Therefore, statins lower blood cholesterol in two ways: they reduce LDL cholesterol (LDL-C) synthesis in cells by inhibiting HMG-CoA reductase, and they increase the number of receptors for low-density lipoproteins (LDL-R) in cells, thereby removing more cholesterol-rich LDL particles from the blood, so that the serum LDL-C level is decreased. It is well known that increased LDL-C is the main risk factor for atherosclerotic cardiovascular disease (ASCVD), particularly coronary heart disease, and that the decrease in elevated LDL-C is essential for the prevention of ASCVD [1]. However, statins do not reduce ASCVD morbidity and mortality only by decreasing LDL-C. They have many other anti-atherosclerotic pleiotropic effects independent of their cholesterol-lowering properties, among which anti-inflammatory effects are quite important [2,3]. These unique properties make statins remaing as the most widely prescribed lipid-lowering medications despite several new drugs have that have been introduced to the market [4,5].

It is well known that atherosclerosis is, among others, a low-grade inflammatory disease and that oxidative stress, including iron-induced oxidative stress, plays an important role in inflammatory responses [6]. Acute-phase reaction is a systemic response that typically occurs at the beginning of an inflammatory process [7]. Ferritin and high-sensitivity C Reactive Protein (hs CRP), as positive acute-phase reactants, promote LDL-particle oxidation and cause inflammation in blood vessels.

Ferritin is involved in immune regulation, and it is important to ensure that iron is safely stored and detoxified, preventing the oxidative damage that could be caused by reactions such as Haber–Weiss or Fenton. Ferritin can be found in serum [8], and if elevated, it is considered to be one of the markers of inflammation [9,10]. It can be an enhancer of the inflammatory response increasing the expression of several proinflammatory mediators [11]. A recently published study on a mouse model showed that ferritin could regulate the progress of atherosclerosis by regulating the expression levels of matrix metalloproteinases and interleukins and that silencing ferritin could inhibit the development of atherosclerosis [12]. Some studies showed that ferritin might be an independent risk factor of arterial stiffness [13,14], and arterial stiffness seems to be an independent marker of ASCVD risk [15].

The presence of a statin–iron nexus is proposed based on data indicating that statins modify iron homeostasis and that both statins and reduced ferritin levels appear to be helpful in lowering oxidative stress and related inflammation, resulting in better clinical outcomes [16]. Increased intracellular ferritin greatly contributes to the cytoprotective action of statin-induced heme oxygenase-1(HO-1), which improves iron mobilization and reduces iron levels in atherosclerotic plaques [17]. However, statins may decrease inflammatory as well as noninflammatory mechanisms that cause acute-phase reactions [17,18].

Several studies were published, mostly during the last decade, presenting conflicting results concerning the effects of statins on serum ferritin levels. Therefore, the primary aim of this systematic review and meta-analysis was to find whether statins do have any effect on serum ferritin levels or not. The second goal was to assess the effect of statins on the levels of another marker of inflammation—hs CRP.

## 2. Methods

### 2.1. Search Strategy

As described by Jamialahmadi et al. [19], this systematic review and meta-analysis was performed in accordance with the 2009 preferred reporting items for systematic reviews and meta-analysis (PRISMA) guidelines [20]. PubMed, Scopus, and Embase, as well as Web of Science, were searched from inception to 5 March 2022, and the following keywords were used in titles and abstracts: (“Hydroxymethylglutaryl-CoA Reductase Inhibitors” OR simvastatin OR rosuvastatin OR atorvastatin OR pravastatin OR pitavastatin OR mevastatin OR fluvastatin OR lovastatin OR cerivastatin) AND (Ferritin OR Ferritins).

### 2.2. Study Selection

Only studies on humans were included, if they were based on the following inclusion criteria: (i) observational studies assessing the effect of statins on ferritin level, regardless of having a control group or not, and (ii) studies with adequate information at baseline and at the end of the study in each group. Exclusion criteria were: (i) lack of sufficient data at baseline or follow-up.

### 2.3. Data Extraction

After the removal of duplicate studies, two independent and blinded authors (T.J. and M.A.) reviewed and screened the titles and abstracts of the eligible studies. Any disagreements were resolved via discussion and consensus. After reviewing the full report of the studies, the following data were abstracted: the name of the first author, the year of publication, the study design, the type and dose of statins, and the duration of follow-up, as well as patient characteristics.

### 2.4. Quality Assessment

The Newcastle–Ottawa Scale (NOS) was applied to evaluate the two cohort studies [21]. Four studies were evaluated using the Risk of Bias in Nonrandomized Studies—on Interventions (ROBINS-I) tool [22]. In addition, Cochrane Collaboration’s tool was used to assess the risk of bias in three randomized studies [23].

### 2.5. Quantitative Data Synthesis

According to a previous study by Jamialahmadi et al. [19], Comprehensive Meta-Analysis (CMA) V2 software (Biostat, NJ, USA) [24] was used to perform the meta-analysis. For each outcome, sample sizes, means, and standard deviations were obtained to calculate the weighted mean differences (WMDs). The effect size was calculated as: (Difference between measured before and after follow-up in the treatment group) − (Difference between measured before and after follow-up in the control group). A random-effects model and the generic inverse-variance weighting method were used to compensate for the heterogeneity of the studies in terms of study design, treatment duration, and the characteristics of the studied populations [20]. In the case of median and range (or 95% confidence interval (CI)), mean and standard deviation were estimated using the method by Hozo et al. [25]. To evaluate the influence of each study on the overall effect size, a sensitivity analysis was performed using the leave-one-out method (i.e., removing one study each time and repeating the analysis) [26,27].

### 2.6. Meta-Regression

As already described in our previous study [19], the potential confounder of the treatment duration of statin treatment was entered into a random-effects meta-regression model to explore their possible interaction with the estimated effect size.

### 2.7. Publication Bias

The presence of publication bias was assessed using funnel plot, Begg’s rank correlation, and Egger’s weighted regression tests in the meta-analysis. When funnel plot asymmetry was seen, potentially missing studies were imputed using the “trim and fill” method. When there was evidence of significant results, the number of potentially missing studies required to make the *p*-value non-significant was estimated using the “fail-safe N” method as another marker of publication bias [28].

## 3. Results

Among the 72 published studies identified by performing a systematic database search, 46 were directly related to the topic of this study after the exclusion of duplicates. However, 37 studies were excluded after careful evaluation (5 were review articles, 22 were irrelevant studies, 3 were excluded because of insufficient data, and 7 were animal studies). Therefore, nine observational studies were included in the systematic review and meta-analysis (Table 1). The study selection process is shown in Figure 1.

### 3.1. Risk-of-Bias Assessment of Clinical Trials

Two cohort studies met the representativeness of Exposed Cohort, Selection of the Non-Exposed Cohort, Ascertainment of Exposure, Comparability of Cohorts on the Basis of the Design or Analysis, and Assessment of Outcome and lacked the presenting of the outcome of interest at the start of study as well as adequacy of follow-up. Among four nonrandomized studies, one study had a critical bias, and three had serious bias. In addition, Cochrane Collaboration’s tool assessed a risk of bias in three randomized studies. All of the selected publications showed a high risk of bias. The quality of the included publications is assessed in Table 2.

### 3.2. Effect of Statins on Circulating Concentrations of Serum Ferritin

The meta-analysis of nine trials including 1611 patients demonstrated a significant decrease in circulating ferritin (WMD, −39.491; 95% CI, −66.56, −12.42; *p* = 0.004) (Figure 2A). The reduction in circulating ferritin because of statin treatment was robust in the leave-one-out sensitivity analyses (Figure 2B).

### 3.3. Meta-Regression

Random-effects meta-regression was performed to assess the impact of potential confounders on the concentrations of the ferritin-lowering effects of statins. The results did not suggest any significant association between the changes in concentrations of ferritin and treatment duration (coefficient, 4.61; 95% CI, −3.55, 12.78; *p* = 0.26) (Figure 3).

### 3.4. Effect of Statins on Circulating Concentrations of Serum hs_CRP

The meta-analysis of six trials [30,31,32,33,35,37] including 1507 patients demonstrated a significant decrease in circulating high-sensitivity CRP (hs_CRP) (SMD, −0.534; 95% CI, −0.950, −0.119; *p* = 0.012) (Figure 4A). The reduction in circulating hs_CRP because of statin treatment as robust in the leave-one-out sensitivity analyses (Figure 4B).

### 3.5. Publication Bias

Given the asymmetric funnel plot, Egger’s linear regression test (intercept = −0.26; standard error = 0.80; 95% CI = −2.07, 1.55; *t* = 0.32; df = 9; two-tailed *p* = 0.75) and Begg’s rank correlation test (Kendall’s Tau with continuity correction = 0.145; *z* = 0.62; two-tailed *p*-value = 0.53) suggested no publication biases in the meta-analysis of the effects of statins on the serum level of ferritin. Using the “trim and fill” method, no potentially missing studies were imputed, yielding an adjusted effect size (WMD) of −39.49 (95% CI: −66.55, −12.42). The “fail-safe N” test showed that 11 missing studies would be needed to bring the effect size down to a non-significant (*p* > 0.05) value (Figure 5).

## 4. Discussion

The results of this meta-analysis, which included nine studies (1611 patients) analyzing the effects of statins on serum ferritin levels, showed a significant decrease in circulating ferritin levels caused by treatment with statins. The results did not suggest any significant association between the changes in concentrations of serum ferritin and the duration of treatment with statins. In addition, hs_CRP levels significantly decreased after treatment with statins. This effect might be explained by the anti-inflammatory pleiotropic effects of statins and not their lipid-lowering effects.

It has to be stressed that a recent population-based cohort study on 242,084 participants who were followed for more than 8 years showed that high levels of serum ferritin were not associated with an increased risk of acute CHD [38]. The negative results were explained by the overall lower prevalence of CHD in the analyzed population compared with other studies. These results are similar to an earlier study on black population (n = 4659) without known atherosclerotic CHD suggesting that higher serum ferritin levels were not associated with an increased risk of atherosclerotic CHD events or stroke [39]. This study did not account for comorbidities such as chronic inflammatory conditions, which can affect blood ferritin levels and were associated with CHD in several earlier studies. Furthermore, because they only included black population, our findings may not be generalizable to other ethnicities or the general public. These two reasons were suggested as controversies in previous studies.

On the other hand, the FeAST trial, a clinical trial of iron reduction by graded phlebotomy, showed that iron reduction caused a significant age-related improvement in ASCVD outcomes. A small substudy of this trial lasting six years that used phlebotomy showed that statin treatment was associated with significantly lower ferritin levels [6]. Another substudy of the FeAST trial also showed results similar to those obtained by our meta-analysis, indicating that statins increased the high-density lipoprotein cholesterol (HDL-C)-to-LDL-C ratios and reduced ferritin levels via noninteracting mechanisms [16,40]. One of the proposed mechanisms connecting statins and iron metabolism, including ferritin levels, is that in secondary prevention, statins may in part stabilize atherosclerotic plaques by inducing intralesional heme oxygenase-1 (HO-1). HO-1 is the rate-limiting enzyme involved in heme catabolism [17] that facilitates iron mobilization and decreases the iron levels in the atherosclerotic plaque [18].

This study has some limitations. A major limitation was the relatively small number of the included studies and the relatively small number of participants in some of the included studies. Another possible limitation might have been the lack of simultaneous measurements and analysis of other markers of inflammation such as interleukin (IL)-1beta, IL-6, IL-10, and tumor necrosis factor (TNF)-alpha.

## 5. Conclusions

This meta-analysis showed that statin therapy caused a significant decrease in circulating ferritin levels. The results did not suggest any significant association between the changes in concentrations of serum ferritin and the duration of treatment with statins. In addition, statin therapy reduced hs_CRP levels in circulation. This effect might be explained by the anti-inflammatory and maybe some other pleiotropic effects of statins and not their lipid-lowering effects.

## Figures and Tables

**Figure 1 jcm-11-05251-f001:**
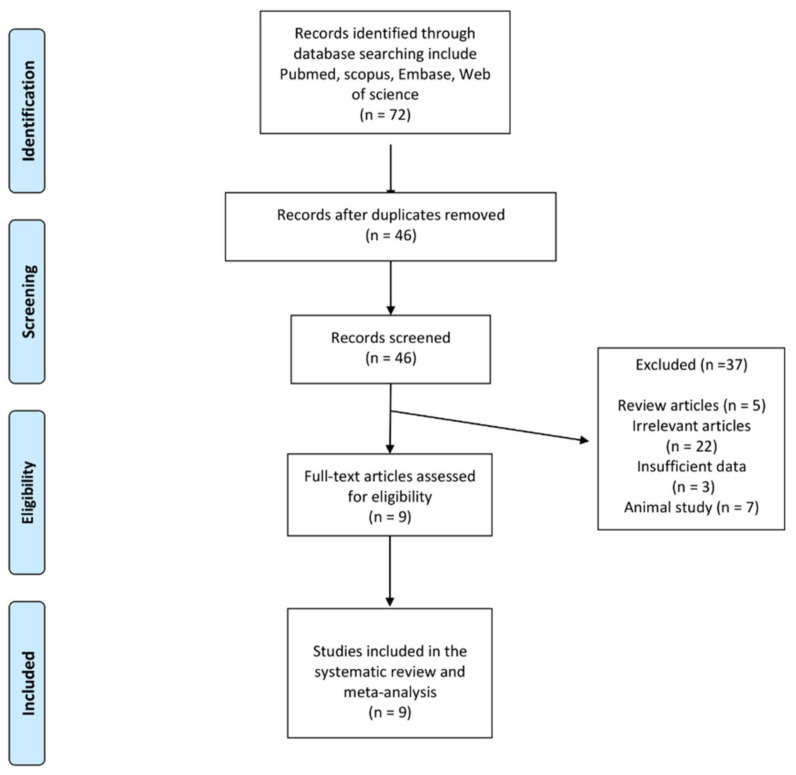
Flow chart of identified publications and those included into the meta-analysis.

**Figure 2 jcm-11-05251-f002:**
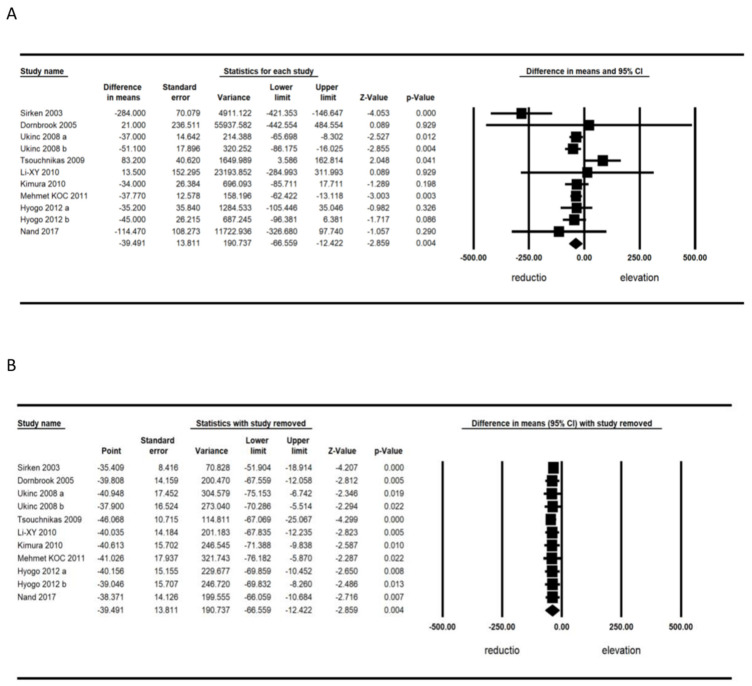
(**A**) Forest plot displaying standardized mean differences and 95% confidence intervals showing the consequence of statin treatment on serum ferritin level. (**B**) Leave-one-out sensitivity analyses [26,27,28,29,30,31,32,33,34].

**Figure 3 jcm-11-05251-f003:**
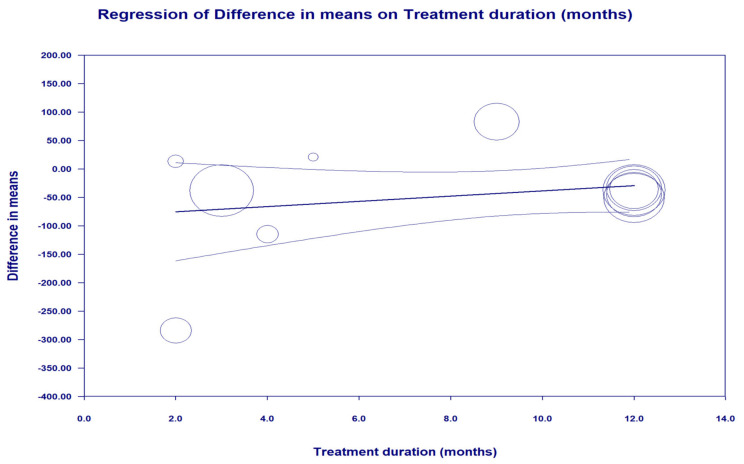
Random-effects meta-regression for evaluating the effect of treatment duration.

**Figure 4 jcm-11-05251-f004:**
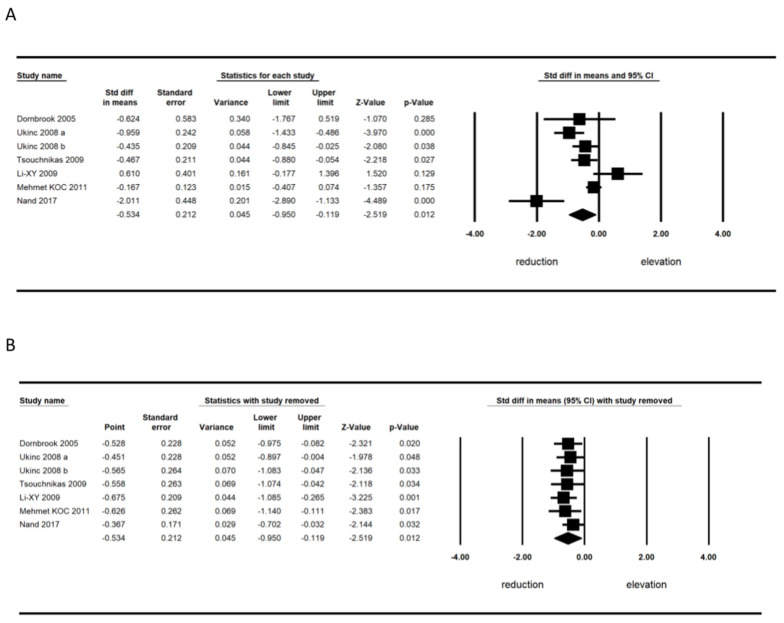
(**A**) Forest plot displaying standardized mean differences and 95% confidence intervals showing the consequence of statin treatment on serum hs_CRP level. (**B**) Leave-one-out sensitivity analyses [27,28,29,30,31,32,33,34].

**Figure 5 jcm-11-05251-f005:**
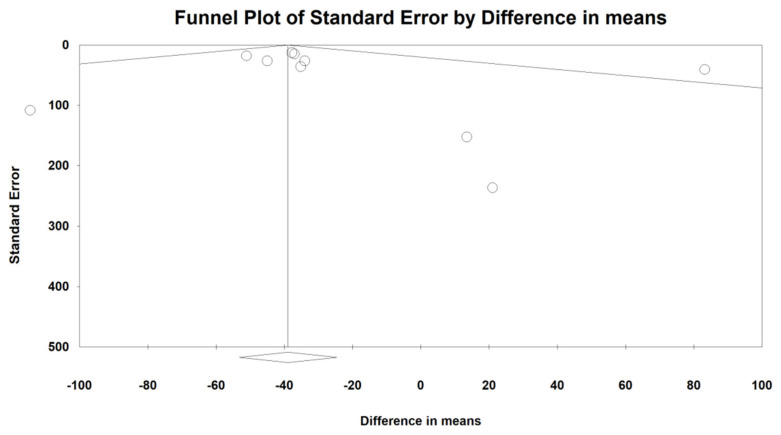
Funnel plot detailing publication bias in the publications describing the effect of statin treatment on serum ferritin level.

**Table 1 jcm-11-05251-t001:** Characteristics of studies in which circulating concentrations of serum ferritin were measured.

Study, Year	Study Design	Follow-Up	Treatment	Control	Clinical Outcome	Patients	No. of Patients
Sirken et al., 2003 [29]	retrospective study	mean of 4.7 months	statin therapy	non-statin therapy	significant decrease in serum ferritin level	dialysis patients	19
Dornbrook-Lavender et al., 2005 [30]	randomized, parallel-group substudy	8 months	A at 10 mg/day	no statins	no significant changes in serum ferritin level	hemodialysis patients	13
Ukinc et al., 2009 [31]	open-label, randomized study	12 months	A at 10 to 20 mg/day	-	significant decrease in serum ferritin level	patients with type 2 diabetes	50
			S at 10 to 20 mg/day	-	significant decrease in serum ferritin level		
Tsouchnikas et al., 2009 [32]	nonrandomized clinical trial	9 months	A at 20 to 40 mg/day	-	no significant changes in serum ferritin level	hemodialysis patients with low-density lipoprotein cholesterol (LDL) >32.58 mmol/L	25
Li et al., 2010 [33]	nonrandomized clinical trial	2 months	S at 20 mg/day	control	no significant changes in serum ferritin level	hemodialysis patients	26
Kimura et al., 2010 [34]	nonrandomized clinical trial	12 months	A at 10 mg/day	-	significant decrease in serum ferritin level	NASH patients with dyslipidemia	43
Mehmet KOC et al., 2011 [35]	retrospective study	3 months	statin	statin nonusers	no significant changes in serum ferritin level	hemodialysis patients	1363
Hyogo et al., 2012 [36]	nonrandomized clinical trial	12 months	A at 10 mg/day	-	no significant changes in serum ferritin level	NASH patients with dyslipidemia (males)	28
					significant change in serum ferritin level	NASH patients with dyslipidemia (females)	14
Nand et al., 2018 [37]	prospective randomized controlled study	4 months	A at 20 mg/day	control	significant change in serum ferritin level	hemodialysis patients	30

A = atorvastatin. S = simvastatin.

**Table 2 jcm-11-05251-t002:** Quality of bias assessment of the included publications.

Study	Selection				Comparability	Exposure		
	Representativeness of the Exposed Cohort	Selection of the Non-Exposed Cohort	Ascertainment of Exposure	Demonstration That Outcome of Interest Was Not Present at Start of Study	Comparability of Cohorts on the Basis of the Design or Analysis	Assessment of Outcome	Was Follow-Up Long Enough for Outcomes to Occur?	Adequacy of Follow-Up of Cohorts
Sirken et al., 2003 [29]	*	*	*	-	*	*	-	-
Mehmet KOC et al., 2011 [35]	*	*	*	-	*	*	-	-
	**Confounding**	**Participant Selection**	**Interventions Classification**	**Intended Interventions Deviations**	**Missing Data**	**Outcome Measurement**	**Reported Results Classification**	
Hyogo et al., 2012 [36]	serious	low	low	low	low	low	low	
Tsouchnikas et al., 2007 [32]	serious	low	low	low	serious	low	low	
Xiang-Yang Li et al., 2010 [33]	low	serious	low	low	serious	low	low	
Kimura et al., 2010 [34]	critical	low	low	low	low	low	low	
	**Selection Bias**		**Performance Bias**		**Detection Bias**	**Attrition Bias**	**Reporting Bias**	**Other Bias**
	**Random Sequence Generation**	**Allocation Concealment**						
Dornbrook-Lavender et al., 2005 [30]	high		high	high	high	low	low	low
Ukinc et al., 2009 [31]	high		high	high	high	low	low	unclear
Nand et al., 2017 [37]	high		high	high	high	low	low	low

## Data Availability

Not applicable.

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
