# Peer review of "The Effects of Statin Treatment on Serum Ferritin Levels: A Systematic Review and Meta-Analysis"

_jcm, 2022, doi:10.3390/jcm11175251_

Round 1

Reviewer 1 Report

The meta analysis performed by Jamialahmadi, et al. is helpful for the clinicians to understand the effect of statins on ferritin. However, there are several critical concerns, especially the number of patients included in the study,  that need to be addressed before sent to publication.

1. Line 42-43, the sentence Therefore, the number of receptors... is not logically fit the context. Is that the number of receptors is compensatory increased because of statins treatment or ASCVD? Please specify.

2. Line55, please define CRP when it firstly appears.

3. In the second paragraph of the introduction, the authors introduced lots of studies to present the correlation between ferritin and inflammation, and highly suggested a therapeutic effect of reducing ferritin in atherosclerosis. It may be OK for this paragraph. However, in the third paragraph, there was none of the introduction of how ferritin may be regulated by statins. What observations and contradictions are there currently in clinical practice concerning the regulation of statins on ferritin? These introductions will make the manuscript more logical and interesting.

4. The relationships between ferritin,CRP and albumin are weak, and there were no introduction of their relationship, so that when the data of CRP and albumin are the major parts of the results section, it is very sudden and strange.

5. The number of patients in the 9 study is 38, 19, 50.....38, respectively. The sum of the numbers is far less than 1611, please explain.

6. Line 172, 6 trials, please specify what are the 6 trials, why the other 3 trials were excluded?

7. And this study focused on ferritin, why CRP and albumin were included? Then why 7 trials were selected for albumin study? What are the 7 trials?

8. Please label A and B correctly in all the figures.

9. In the figs.2, 4, and 5, I supposed that Ukinc 2008a and Ukinc 2008b were from the same study, and Hyogo 2012a and Hyogo 2012b were from the same study. Please clarify. In addition, the years presented in the figs didnt match that in the table 1, please explain.

10. In the discussion, the authors introduced two studies whose results does not consistent with the conclusion of the article. It is important for the readers to know the information. However, the authors should explain and discuss this contradiction not only listed them.

Author Response

Dear Editor,

Thank you for the comments concerning our manuscript entitled “The effects of statin treatment on serum ferritin levels: a systematic review and meta-analysis”. We appreciate very much the reviews, constructive comments, and suggestions to our manuscript. We have reviewed the manuscript carefully and we have revised it accordingly. Our responses are given point-by-point below. The suggested changes are highlighted in yellow in the revised manuscript. Since we have made all the suggested changes, we hope that you will accept the revised manuscript for publication.

Best regards

Amirhossein Sahebkar, Pharm.D, Ph.D, Fellowship
Biotechnology Research Center,

Pharmaceutical Technology Institute,

School of Pharmacy,

Department of Medical Biotechnology,

School of Medicine,
Mashhad University of Medical Sciences
Mashhad 9177948564, Iran
Tel: +98-513-8002299
Cell: +98-915-1221496
Fax: +98-513-8002287
E-mail: 
sahebkara@mums.ac.ir

amir_saheb2000@yahoo.com

Highly cited researcher in the field of Pharmacology & Toxicology (Essential Science Indicator top 1% global list)

Highly cited researcher in the field of Clinical Medicine (Essential Science Indicator top 1% global list)

Highly cited researcher in the field of Biology & Biochemistry (Essential Science Indicator top 1% global list)

Highly cited researcher in the field of Molecular Biology & Genetics (Essential Science Indicator top 1% global list)

Highly cited researcher in the field of Agricultural Sciences (Essential Science Indicator top 1% global list)

h-index: 119

#1 Reviewer:

The meta analysis performed by Jamialahmadi, et al. is helpful for the clinicians to understand the effect of statins on ferritin. However, there are several critical concerns, especially the number of patients included in the study,  that need to be addressed before sent to publication.

Comment 1: Line 42-43, the sentence “Therefore, the number of receptors...” is not logically fit the context. Is that the number of receptors is compensatory increased because of statins treatment or ASCVD? Please specify.

Response: Thank you for your kind first sentence. We agree with your comment 1. and the sentence was rewritten to be more clear.

Comment 2: Line55, please define “CRP” when it firstly appears.

Response: Thank you for your comment. The “CRP” definition has been added.

Comment 3: In the second paragraph of the introduction, the authors introduced lots of studies to present the correlation between ferritin and inflammation, and highly suggested a therapeutic effect of reducing ferritin in atherosclerosis. It may be OK for this paragraph. However, in the third paragraph, there was none of the introduction of how ferritin may be regulated by statins. What observations and contradictions are there currently in clinical practice concerning the regulation of statins on ferritin? These introductions will make the manuscript more logical and interesting.

Response: We have incorporated reviewer’s suggestion in “Introduction” section discussing the  suggested topic.

Comment 4: The relationships between ferritin, CRP and albumin are weak, and there were no introduction of their relationship, so that when the data of CRP and albumin are the major parts of the results’ section, it is very sudden and strange.

Response: We agree with the reviewers remark and therefore the discussion on albumin was accordingly removed to ensure more clarity and better flow of the text.

Comment 5: The number of patients in the 9 study is 38, 19, 50.....38, respectively. The sum of the numbers is far less than 1611, please explain.

Response: Thank you for pointing this out. The number of patients in Table 1 which was written by a mistake, is now corrected.

Comment 6: Line 172, 6 trials, please specify what are the 6 trials, why the other 3 trials were excluded?

Response: Thank you very much for your comment. Only 6 trials having the data on hs_CRP levels were cited accordingly.

Comment 7: And this study focused on ferritin, why CRP and albumin were included? Then why 7 trials were selected for albumin study? What are the 7 trials?

Response: Thank you for your comment. Discussion on the association of albumin with statins was removed but we included hs CRP as secondary endpoint.

Comment 8: Please label A and B correctly in all the figures.

Response: The change has been made accordingly.

Comment 9: In the figs.2, 4, and 5, I supposed that Ukinc 2008a and Ukinc 2008b were from the same study, and Hyogo 2012a and Hyogo 2012b were from the same study. Please clarify. In addition, the years presented in the figs didn’t match that in the table 1, please explain.

Response: As suggested by the reviewer, we double-checked the publications’ years and changed them when necessary. Patients in Ukinc and Hyogo publications were divided into two subgroups which were named as “a” and “b”.

Comment 9: In the discussion, the authors introduced two studies whose results does not consistent with the conclusion of the article. It is important for the readers to know the information. However, the authors should explain and discuss this contradiction not only listed them.

Response: As suggested by the reviewer, we have explained this controversy in “Discussion” section.

Reviewer 2 Report

This is a well-prepared systematic review dealing with an important topic. But, some concerns need to be addressed to fit for publication as follows:

1. Keywords: add more keywords like statin; cardiovascular disease.

2. Line 70-71: the authors begin the paragraph with some studies, then end with one reference (13). Please, add the references of some studies.

3. The discussion section is the major concern of this review. The discussion is very concise. I would recommend the authors give more discussions on lessons learned from the state of the science and challenges in this field in the discussion section to show the manuscript's contribution more clearly. 

4. Conclusion is very concise and needs to be supported with further perspectives.

5. There is a problem with using abbreviations throughout the manuscript. The full term should be mentioned first with the abbreviation between paresis then the abbreviations should be exclusively used throughout the manuscript. E.g., in line 55, CRP should be written as C- reactive protein (CRP), then the abbreviation should be exclusively used further. Such errors have been repeated for many abbreviations throughout the manuscript.

Author Response

Dear Editor,

Thank you for the comments concerning our manuscript entitled “The effects of statin treatment on serum ferritin levels: a systematic review and meta-analysis”. We appreciate very much the reviews, constructive comments, and suggestions to our manuscript. We have reviewed the manuscript carefully and we have revised it accordingly. Our responses are given point-by-point below. The suggested changes are highlighted in yellow in the revised manuscript. Since we have made all the suggested changes, we hope that you will accept the revised manuscript for publication.

Best regards

Amirhossein Sahebkar, Pharm.D, Ph.D, Fellowship
Biotechnology Research Center,

Pharmaceutical Technology Institute,

School of Pharmacy,

Department of Medical Biotechnology,

School of Medicine,
Mashhad University of Medical Sciences
Mashhad 9177948564, Iran
Tel: +98-513-8002299
Cell: +98-915-1221496
Fax: +98-513-8002287
E-mail: 
sahebkara@mums.ac.ir

amir_saheb2000@yahoo.com

Highly cited researcher in the field of Pharmacology & Toxicology (Essential Science Indicator top 1% global list)

Highly cited researcher in the field of Clinical Medicine (Essential Science Indicator top 1% global list)

Highly cited researcher in the field of Biology & Biochemistry (Essential Science Indicator top 1% global list)

Highly cited researcher in the field of Molecular Biology & Genetics (Essential Science Indicator top 1% global list)

Highly cited researcher in the field of Agricultural Sciences (Essential Science Indicator top 1% global list)

h-index: 119

2 Reviewer:

This is a well-prepared systematic review dealing with an important topic. But, some concerns need to be addressed to fit for publication as follows:

Comment 1: Keywords: add more keywords like statins; cardiovascular disease...

Response: Thank you for the kind first sentence. We have added the suggested keywords.

Comment 2: Line 70-71: the authors begin the paragraph with some studies, then end with one reference (13). Please, add the references of some studies...

Response: We have included the relevant reference at the end of the sentence.

Comment 3: The discussion section is the major concern of this review. The discussion is very concise. I would recommend the authors give more discussions on lessons learned from the state of the science and challenges in this field in the discussion section to show the manuscript's contribution more clearly.

Response: We have gone through the entire manuscript carefully, particularly through the Discussion,  and have discussed more challenges accordingly.

Comment 4: Conclusion is very concise and needs to be supported with further perspectives...

Response: The changes have been made as suggested by the reviewer.

Comment 5: There is a problem with using abbreviations throughout the manuscript. The full term should be mentioned first with the abbreviation between paresis then the abbreviations should be exclusively used throughout the manuscript. E.g., in line 55, CRP should be written as C- reactive protein (CRP), then the abbreviation should be exclusively used further. Such errors have been repeated for many abbreviations throughout the manuscript...

Response: We have incorporated your suggestion throughout the paper.

Round 2

Reviewer 1 Report

The manuscript has been improved greatly. I have no further suggestions.

Reviewer 2 Report

No further comments to be addressed